# Conditional Idempotent Generative Networks

## Abstract

We propose Conditional Idempotent Generative Networks (CIGN), a new approach that expands upon Idempotent Generative Networks (IGN) to enable conditional generation. While IGNs offer efficient single-pass generation, they lack the ability to control the content of the generated data. CIGNs address this limitation by incorporating conditioning mechanisms, allowing users to steer the generation process towards specific types of data.

We establish the theoretical foundations for CIGNs, outlining their scope, loss function and evaluation metrics. We then present two potential architectures for implementing CIGNs, which we call channel conditioning and filter conditioning. We discuss experimental results obtained on the MNIST dataset, demonstrating the effectiveness of both conditioning approaches. Our findings pave the way for further exploration of CIGNs on larger datasets and more complex use cases.

## 1 Introduction

In the seminal *Idempotent Generative Network* paper Shocher et al. (2023), the authors propose a novel approach to generation which they call Idempotent Generative Network (IGN). One major advantage of this approach compared to diffusion (or autoregressive) models is that generation is done in a single forward pass, and is therefore several times faster.

However, the paper (Shocher et al., 2023) does not touch into how to *condition* that generation: a well-trained idempotent generative network will generate synthetic data resembling examples seen during training, but the authors do not describe how to nudge the IGN towards generating certain examples (for example, examples belonging to a certain class) over others.

Our paper starts answering that question: we describe how to expand the theory of idempotent generative networks to include a conditioning mechanism, and we call these networks *Conditional Idempotent Generative Networks*. We further describe two possible architectures for implementing conditional idempotent generative networks, and we discuss experimental results.

Our main contributions are then:

1. We set the theoretical foundations for Conditional Idempotent Generative Networks, describing their scope and intended use, the format of the loss function and what metrics can be used to evaluate them.

2. We describe two possible implementations of the conditioning mechanism of a Conditional Idempotent Generative Network - we call these mechanisms *channel conditioning* and *filter conditioning*.

3. We discuss results of our experiments comparing the two implementations of CIGN on the MNIST dataset.

Our experiments show that both channel conditioning and filter conditioning are effective approaches in implementing a Conditional Idempotent Generative Networks. Further experiments on larger datasets and with more powerful computing power are needed to identify which mechanism is more effective.

Our code is available at
`https://github.com/niccronc/conditional-idempotent-generative-networks`.

## 2 Idempotent generative networks

We start by summarizing the ideas leading to the concept of idempotent generative networks, after Shocher et al. (2023).

The authors start from the observation that typically the true data distribution lives in a low-dimensional submanifold $D$ of the feature space $X$. *Generating* a synthetic data point corresponds then to choosing a point on this low-dimensional submanifold $D$. One possible way to choose a point on this low-dimensional submanifold $D$ is to have a function $F$ whose image is $D$, and then apply $F$ to a random input.

The authors' idea is to construct $F$ to be an idempotent function[1] on the feature space, with image constrained to be *as close as possible* to $D$.

A machine learning network (called *idempotent generative network*) is then trained to learn the parameters of the idempotent function $F : X \to X$ with image $D$.

The difficulty of creating synthetic data points (in other words - the difficulty of describing an accurate parametrization of the submanifold $D$ starting from a training dataset) is then subsumed into the difficulty of identifying optimal parameter for the function $F$. A well-trained idempotent generative network $F$ will then generate data points $F(x)$ when applied to any randomly selected point $x$ in the feature space $X$. In particular, generation happens in a single forward pass and is therefore considerably faster than autoregressive or diffusion generative methods.

We summarize the benefits of using idempotent generative networks:

- Single-step generation: unlike diffusion or autoregressive models, IGNs generate outputs in one step.

- Optional refinements: similar to diffusion models, IGNs allow for optional sequential refinements (meaning: we can pass the output through the network a second time, and since the trained $\hat{F}$ is not truly idempotent, we may obtain a slightly higher quality output - this was discussed in Shocher et al. (2023)).

- No feature/latent space conundrum: unlike diffusion models, IGNs work only on the feature space, facilitating manipulations and interpolations of inputs and outputs.

- Output enhancement: an IGN could be used to project degraded data back onto the original data distribution.

### 2.1 Loss function

While the initial version of the paper Shocher et al. (2023) does not provide insights on the model architectures used for the experiments the authors describe, the paper does discuss at length the loss function of an IGN from a theoretical perspective.

Let again $X$ be the feature space and $D$ be the data manifold. Let $F : X \to X$ be the idempotent function we are trying to construct and whose parameters are being learned by the IGN.

The loss function is then composed of the following three terms:

- (reconstruction) Real data samples $d$ from the data distribution $D$ should remain unchanged by the model: $F(d) = d$.

- (idempotent) Any further application of the model beyond the first one should act like the identity function: $F(F(z)) = F(z)$ for every $z$ in $X$.

---

[1] A function $f : X \to X$ is called idempotent if, when applied multiple times, it produces the same output as applying it once: $(f(f(x)) = f(x))$.

- (tightness) The previous two conditions should be satisfied while ensuring $F$ has as small a range as possible.

We encourage the interested reader to check out additional details in the original paper Shocher et al. (2023).

## 2.2 Network architecture

The theoretical framework of idempotent generative network is fundamentally architecture-agnostic. And yet, one needs to choose a model architecture in order to test the feasibility of the idea in practice.

In Shocher et al. (2023), the authors do not explicitly describe the architecture used in their experiments. Inspired by Pulfer (2024), we will leverage a GAN-like architecture.

In the classic setup of Generative Adversarial Networks, a generator and a discriminator compete in a zero-sum game: the generator creates synthetic data points resembling as much as possible a real dataset, while the discriminator attempts to distinguish the real data points from the synthetic ones. In particular, in the GAN framework the discriminator is fed synthetic data points created by the generator.

In the IGN setting, instead, the two models will work in the opposite order: the discriminator $d$ processes input data into a latent tensor, and the generator $g$ creates a synthetic data point from that latent tensor. Our idempotent generative network is then $F = g \circ d$.

## 3 Conditioning idempotent generative networks

Idempotent generative networks are then both simple and elegant from a theoretical perspective, as well as powerful from a practical one: in particular IGNs can generate data in a single step, without the multi-steps denoising process implemented by diffusion models or the 'single-token-generation' approach of autoregressive text generation models.

However, the approach taken in Shocher et al. (2023) leaves one crucial question unanswered: can we condition the model to generate new samples with prescribed characteristics?

We present multiple approaches to answer affirmatively the previous question.

### 3.1 Theoretical perspective

Let $X$ be again the feature space and $D$ be the data manifold, immersed into $X$. Suppose that every data point in $D$ comes with a *condition*. This could for example be a label or a caption (in case $D$ represent images). Let $C$ be the 'condition' space, the space where conditions $c$ belong to.

We can now extend the paradigm behind idempotent generative network to consider conditioning: we have an augmented data manifold $\widetilde{D}$ immersed in $X \times C$, every point $(d, c)$ consisting of a data point $d$ and its condition $c$. We want to find an idempotent map

$$F : X \times C \to X \times C$$

such that:

- (reconstruction) $F(d, c) = (d, c)$ for every augmented data point $(d, c) \in \widetilde{D}$.

- (idempotence) $F(F(x, c)) = F(x, c)$ for every $(x, c)$ in $X \times C$.

- (tightness) the range of $F$ is *as small as possible*, i.e. $F(X \times C)$ is a low-dimensional submanifold of $X \times C$.

As explained in Shocher et al. (2023), the *tightness* condition is crucial to avoid the model learning to replicate the identity function (which satisfies the *idempotence* and the *reconstruction* requirements).

We plan on using the model as follows: suppose we want to generate a sample with condition $c$. We randomly select a noisy sample $z \in X$, we calculate $F(z, c) \in X \times C$ and we extract the first component $\tilde{d}$ of $F(z, c)$.

### 3.2 Loss function

Even a perfect model $F$ satisfying the three conditions in section 3.1 above is not guaranteed to return a data point $\tilde{d}$ related to condition $c$. Indeed, the idempotent function $F$ is not being explicitly constrained to return a data point $\tilde{d}$ related to condition $c$, just a data point in the extended data manifold.

In order to help the model learn the conditioning, we treat *mismatched* data points like noisy examples. A *mismatched* data point is a pair $(d, \hat{c})$ where $(d, c)$ belongs to the augmented data manifold $\widetilde{D}$ and $c \neq \hat{c}$. In other words, we tell the model that $(d, \hat{c})$ should not be part of the range. By treating mismatched data points like noisy samples, we teach the network to *not* act like the identity function on them.

Let $D_c = \left\{ d \in D \,|\, (d, c) \in \widetilde{D} \right\}$ be the subset of the data manifold $D$ consisting of points with condition $c$. We are then teaching the model $F$ to have range contained in $\widetilde{D}$ (tightness and reconstruction conditions), but that $D_c \times \hat{c}$ should not be part of the range for any $\hat{c} \neq c$ (mismatched conditions). By exclusion this forces the model $F$ to further restrict its range to the unions of $D_c \times c$ as $c$ varies in $C$.

The loss function for conditional idempotent generative networks is then made of five terms:

1. (reconstruction loss $L_{rec}$) Real data samples $(d, c)$ from the augmented data manifold $\widetilde{D}$ should remain unchanged by the model: $F(d, c) = (d, c)$.

2. (idempotent on noisy samples $L_{idem}^{noise}$) For any noisy sample $z$ and any condition $c$, any further application of the model beyond the first one should act like the identity function: $F(F(z, c)) = F(z, c)$.

3. (tightness on noisy samples $L_{tight}^{noise}$) The idempotence condition on noisy samples and the reconstruction condition should be satisfied while ensuring $F$ has as small a range as possible.

4. (idempotent on mismatched samples $L_{idem}^{mism}$) For a data point $(d, c)$ on the augmented data manifolds $\widetilde{D}$ and a condition $\hat{c}$ different from $c$, any further application of the model beyond the first one should act like the identity function: $F(F(d, \hat{c})) = F(d, \hat{c})$.

5. (tightness on mismatched samples $L_{tight}^{mism}$) The idempotence condition on mismatched samples and the reconstruction condition should be satisfied while ensuring $F$ has as small a range as possible.

Each one of these conditions is implemented via the choice of a distance function $\delta$ on the space $X \times C$, which is used to compare a data point before and after the application of $F$.[2] For example the reconstruction loss is implemented as

$$L_{rec}(d, c) = \delta\left((d, c), F(d, c)\right) \quad \forall (d, c) \in \widetilde{D}.$$

The total loss is then obtained as a weighted sum of these five terms:

$$L = w_{rec} \cdot L_{rec} + w_{idem}^{noise} \cdot L_{idem}^{noise} + w_{tight}^{noise} \cdot L_{tight}^{noise} + w_{idem}^{mism} \cdot L_{idem}^{mism} + w_{tight}^{mism} \cdot L_{tight}^{mism}.$$

Each one of the five weights is a hyperparameter that can be tuned during training.

During training, every batch will then be composed of three types of data points:

- true data points $(d, c)$ in the augmented data manifold $\widetilde{D}$.

- noisy samples $(z, c)$ where $z$ is a randomly sampled from $X$ following a prior distribution.

- mismatched samples $(d, \hat{c})$ for $(d, c) \in \widetilde{D}$ and $\hat{c} \neq c$.

The relative proportion of the three types of data points within a batch is a training hyperparameter.

---

[2]For simplicity one usually picks the same distance function $\delta$ for all loss terms, although this is not theoretically necessary.

### 3.3 Metrics

We discuss some theoretical considerations regarding what metrics should be used to evaluate Conditional Generative Idempotent Networks.

At a high level, an appropriate metric should measure the following two dimensions:

- (realism) The synthetic data point $F(z, c)$ generated from condition $c$ should indeed look like a sample from $D_c$.

- (diversity) Synthetic data points $F(z, c)$ generated from the same condition $c$ as we vary the noisy input $z$ should space over as large a subset of $X$ as possible.

There is a fairly standard set of metrics typically used to measure these two dimensions for Generative Adversarial Networks, for example Inception Score (IS) and Fréchet Inception Distance (FID). These metrics have been adapted to the setup of conditional generative adversarial networks (see for example Benny et al. (2021)).

More complex metrics (see for example Ku et al. (2023)) have been suggested for conditional image generation models, especially if the generation is implemented via a diffusion process. In this latter scenario, human evaluation is most of the time still the most accurate performance metric.

All of these metrics are reasonable choices and the one to be preferred mostly depends on the specific use case. If the condition space $C$ is a finite set of classes, metrics based on the inception score and the Fréchet inception distance are probably adequate. If the condition space $C$ is a continuous manifold (for example, $c$ can be any English sentence), then human evaluation is likely to be preferred.

## 4 Two proposed implementations of CIGN

In this section, we described two possible implementations of the Conditional Idempotent Generative Networks.

For both implementations, we assume that the data has a channel dimension (for example audio, images or video data).

Let $(x, c)$ be an element in $X \times C$. We denote by $(n, S)$ the shape of the tensor $x$, where $n$ is the channel dimension and $S = (s_1, \ldots, s_k)$ represents any number of additional dimensions (for example, if $x$ is an image then $S = (h, w)$ is a pair of height and width dimensions).

### 4.1 Channel conditioning

Let $E : C \to \mathbb{R}^S$ be an embedding function, where we denote by $\mathbb{R}^S = \mathbb{R}^{s_1} \times \ldots \times \mathbb{R}^{s_k}$ the subspace corresponding to the non-channel dimensions of the feature space $X$.

*Channel conditioning* is implemented by concatenating $E(c)$ and $x$ along the channel dimension.

This is a very generic and flexible idea, with many different variations available:

- We can build different embeddings $E_i(c)$ whose target space consists of the non-channel dimensions of the input $x_i$ to layer $i$ of the network, and then concatenate $E_i(c)$ to $x_i$.

- We can build embeddings whose target space is multiple copies of $\mathbb{R}^S$, so that when concatenating along the channel diemnsion we give the model additional flexibility on learning from the condition.

In section 5.3 we describe our experiments on training a conditional idempotent generative network with channel conditioning on the MNIST dataset.

### 4.2 Filter conditioning

Let $S' = (s'_1, \ldots, s'_k)$ be any $k$-tuple with $0 < s_i$ for any $1 \leq i \leq k$. Let $E : C \to \mathbb{R}^{S'}$ be an embedding function.

*Filter conditioning* is implemented by calculating cross correlation (or transposed cross correlation) between $x$ and $E(c)$ along the non-channel dimensions. This is inspired by the concept of correlation filters for object detection (see for example Bertinetto et al. (2016)).

One can think of filter conditioning as replacing a standard or transposed convolutional layer (where the model learns the kernel's weights and those weights are shared across all inputs) by a simpler cross-correlation layer, where each input is (transpose) cross-correlated to its condition's embedding, and the model learns the embedding layer's weights.

This also is a very generic and flexible idea, with many different variations available:

- We can choose different filter dimensions $S'$ for the embedding's target space - in fact, $S'$ can be considered as a hyperparameter of the model.

- We can replace any number of convolutional layers (of a base model architecture that we start with) with filter conditioning layers where the filter dimensions $S'$ are the dimensions of the original convolutional layer.

- We can concatenate the result of a standard convolutional layer with the result of a filter conditioning layer, assuming that two layers' parameters (kernel's shape, padding, stride, dilation) are identical.

## 5 Experiments on the MNIST dataset

In this section we discuss our experiments implementing CIGNs on the MNIST dataset. All experiments were run on consumer hardware - specifically one single free GPU offered by Sagemaker Studio Lab, with a maximum session time of 4 hours.

While this severely limited our experiments (in terms of the model size and in terms of the number of hyperparameter runs we experimented with), we are still able to successfully train well-performing Conditional Idempotent Generative Networks (CIGN), able to generate different good quality images from the *same* noisy data point when prompted with different conditions.

### 5.1 Preprocessing steps and high-level architecture

The MNIST dataset from TorchVision consists of images with a single channel and a size of $28 \times 28$ pixels, valued in $[0, 1]$. We preprocess the dataset by transforming images into torch tensors, and then linearly rescaling pixels to be valued between $-1$ and $1$. We add some small random noise to the training data, to fight overfitting.

In the setup of the previous sections, we have then the feature space

$$X = \mathbb{R}^1 \times \mathbb{R}^{28} \times \mathbb{R}^{28},$$

with the condition space

$$C = \{0, ..., 9\}$$

being the label set of the MNIST dataset.

Noisy data points are sampled from $X$ according to a normal distribution centered at 0 and with standard deviation 1, and then clipped so that they belong to $[-1, 1]^{1 \times 28 \times 28}$.

The distance function $\delta$ used to implement the loss function is

$$\delta\big((x_1, c_1), (x_2, c_2)\big) = L^1\big(x_1 - x_2\big) = |x_1 - x_2|_1\,,$$

the $L^1$ norm of the difference between the feature space components. While this is not strictly speaking a distance on $X \times C$, it works for our purposes since the models we implement pass through the condition $c$ 'as-is' (meaning that $F(x, c) = (F_1(x, c), c)$), and for the purpose of loss calculation we only need to calculate the distance between $(x, c)$ and $F(x, c)$.

At a high level, the architecture of our Conditional Idempotent Generative Networks is composed of:

- a discriminator with domain $X \times C$, outputting a pair of a latent tensor and the condition $c$;

- a generator, with input the latent tensor and the condition $c$, and range $X \times C$.

The CIGN is then obtained by running the discriminator first and the generator second.

The architectures of the discriminator and the generator are heavily inspired by the DCGAN framework, in its pytorch implementation ((Pytorch)).

The discriminator is defined as a sequence of convolutional layers with:

- an increasing number of channels, starting from a latent dimension $l_{dim}$ and doubling each layer, with the final latent tensor having size $(i_{dim}, 1, 1)$.[3]

- decreasing non-channel dimensions, until the latent tensor has non-channel dimensions $(1, 1)$ - that is to say, it is a 1-dimensional tensor.

The generator is defined by mirroring exactly the layers of the discriminator, but in the opposite order: a sequence of transpose convolutional layers with decreasing number of channels and increasing non-channel dimensions. The parameters of each convolutional layer of the discriminator (kernel size, stide, padding, dilation) are equal to those of the corresponding transpose convolutional layer of the generator.

## 5.2 Metrics

Our main metric is the average Fréchet Inception Distance across all 10 classes. We remark that this is identical to the metric named WCFID in section 3.2 of Benny et al. (2021).

For each class, we compare the 1000 samples available in the validation dataset from TorchVision to 1000 synthetic samples generated by the model under consideration.

This comparison is implemented via the pytorch-fid library ((Seitzer, 2020)): this library leverages the ImageNet-v3 model to get intermediate embeddings of the images, which are used to estimate the parameters of the data distribution under a normality assumption.

The aforementioned implementation Seitzer (2020) allows the user to choose among four different embeddings (each one passing the image through a partial inception-v3 model (Szegedy et al., 2015) up to a certain layer), but one of them is not available to us due to the small size of the distributions we are comparing (just 1000 samples). The other three embeddings output respectively a 64-dimensional tensor, a 192-dimensional tensor and a 768-dimensional tensor. We denote the corresponding metrics by $\text{FID}_{64}$, $\text{FID}_{192}$ and $\text{FID}_{768}$.

## 5.3 Channel conditioning

We implement the channel conditioning approach from section 4.1 in its simplest form:

1. we train one shared embedding layer

$$E : C \to \mathbb{R}^{\text{emb}_{\text{dim}}},$$

---

[3]We denote by $i_{dim}$ the intermediate dimension. We call this *intermediate* because this is obtained exactly halfway through the forward pass of the entire CIGN, when the latent tensor is outputted by the discriminator and passed to the generator.

2. and then for each (possibly transpose) convolutional layer where the input $x_i$ has non-channel dimensions $(h_i, w_i)$, we train one linear layer

$$L_i : \mathbb{R}^{\text{emb}_{\text{dim}}} \to \mathbb{R}^{h_i \cdot w_i},$$

and we concatenate $L_i(E(c))$ to $x_i$ along the channel dimension, before the forward pass of the (possibly transpose) convolutional layer.

### 5.3.1 Training details

We run two channel conditioning experiments where the models have different hyperparameters, as in table 1.

Table 1: Parameters for channel conditioning experiments

| Model size | Embedding dim | Latent dim | Intermediate dim | Num parameters |
|---|---|---|---|---|
| small | 5 | 32 | 128 | 1.55m |
| large | 5 | 64 | 128 | 5.6m |

For each real data point in the augmented data manifold, we create nine mismatched data points by pairing the image with all other labels except the correct one. The number of noisy images in each batch is equal to the number of real data points. Our proportion of real:noise:mismatched data points is then 1:1:9.

We train the model for 50 epochs, with a learning rate of 0.001 and a batch size of 128 (this means that each backward pass is actually calculated on $128 \times 11 = 1408$ data points, due to the 1:1:9 ratio mentioned in the previous paragraph).

We experiment with a few different choices of weights for the five summands of the loss function described in 3.2, and ultimately we choose the following values:

- $w_{rec} = 20.0$

- $w_{idem}^{noise} = 20.0$

- $w_{tight}^{noise} = 2.5$

- $w_{idem}^{mism} = 3.0$

- $w_{tight}^{mism} = 1.0$

The detailed architecture of generator and discriminator are described in appendix A.1, including figures 1 and 2.

### 5.4 Filter conditioning

We implement the filter conditioning approach from section 4.2 in its simplest form:

1. we train one shared embedding layer

$$E : C \to \mathbb{R}^{\text{emb}_{\text{dim}}},$$

2. for each (possibly transpose) convolutional layer where the standard DCGAN architecture in Pytorch has input $x_i$ with $c_i$ channels and a kernel of size $(h_i, w_i)$, we train one linear layer

$$L_i : \mathbb{R}^{\text{emb}_{\text{dim}}} \to \mathbb{R}^{c_i \cdot h_i \cdot w_i},$$

3. we calculate channel-wise (possibly transpose) cross-correlation between $x_i$ and $L_i(E(c))$,

4. for each (possibly transpose) convolutional layer where the standard DCGAN architecture has $c_{in}$ input channels and $c_{out}$ output channels, we train a *channel mixer* linear layer

$$C_i : \mathbb{R}^{c_{in}} \to \mathbb{R}^{c_{out}},$$

which we apply to the output of the cross-correlation from the previous step.

### 5.4.1 Training details

We run two filter conditioning experiments where the models have different hyperparameters, as in table 2.

Table 2: Parameters for channel conditioning experiments

| Model size | Embedding dim | Latent dim | Intermediate dim | Num parameters |
|---|---|---|---|---|
| small | 5 | 64 | 128 | 616k |
| large | 5 | 96 | 256 | 1.38m |

For each real data point in the augmented data manifold, we create nine mismatched data points by pairing the image with all other labels except the correct one. The number of noisy images in each batch is equal to the number of real data points. Our proportion of real:noise:mismatched data points is then 1:1:9.

We train the model for 50 epochs, with a learning rate of 0.0001 and a batch size of 64 (this means that each backward pass is actually calculated on $64 \times 11 = 704$ data points, due to the 1:1:9 ratio mentioned in the previous paragraph).

We experiment with a few different choices of weights for the five summands of the loss function described in 3.2, and ultimately we choose the following values:

- $w_{rec} = 20.0$

- $w_{idem}^{noise} = 20.0$

- $w_{tight}^{noise} = 2.5$

- $w_{idem}^{mism} = 8.0$

- $w_{tight}^{mism} = 1.0$

The detailed architecture of generator and discriminator is described in appendix A.2, including figures 3 and 4.

### 5.5 Results

In appendices B.1 and B.2 we display samples of the images generated by the best experiments.

In this section we detail the metrics calculated on our trained experiments.

As discussed in section 5.2, our metric is the average Fréchet Inception Distance (FID) across the ten classes. For each trained experiment and each class, we compare the 1000 samples on the MNIST validation dataset with 1000 samples generated by the trained experiment. We also report in table 3 minimum and maximum FID across all ten classes.

Essentially all metrics point to the large experiment trained with channel conditioning as the best experiment of the four. On the other hand, that experiment has about 4x as many trainable parameters as the next two experiments (the small experiment trained with channel-conditioning and the large experiment trained with filter conditioning).

Table 3: FID metrics for all experiments

|  |  | filter_large | filter_small | channel_large | channel_small |
|---|---|---|---|---|---|
| $\text{FID}_{64}$ | mean | 0.8278 | 0.3601 | **0.1149** | 0.6586 |
|  | min | 0.0837 | 0.0959 | **0.0455** | 0.1126 |
|  | max | 3.25 | 0.7516 | **0.272** | 2.1216 |
| $\text{FID}_{192}$ | mean | 3.3095 | 1.792 | **0.6598** | 2.7955 |
|  | min | 0.4912 | 0.5992 | **0.3735** | 0.5908 |
|  | max | 12.5165 | 5.1655 | **1.1886** | 8.2649 |
| $\text{FID}_{768}$ | mean | 0.4423 | 0.3384 | **0.2852** | 0.4619 |
|  | min | 0.2485 | 0.1869 | **0.1576** | 0.3557 |
|  | max | 0.8765 | **0.4641** | 0.5184 | 0.7127 |

The metrics for the small experiment with filter conditioning (the smallest model of all, with only 616k trainable parameters) are clearly better than the small experiment with channel conditioning.

The metrics for the large experiment with filter conditioning are unusually off. Human review of the synthetic images generated by this model suggests that it tends to create 'thick font' images (see appendix B.2) which is likely the reason for the large FID values and therefore poor results. We have not investigated why this experiment tends to generate 'thick font' images.

In conclusion, we believe that the jury is still out regarding which of the conditioning mechanism is better - the best performing experiment is the large model with channel conditioning, but the small filter conditioning model clearly outperforms a slightly larger model trained with channel conditioning.

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

## A   Model architectures

In this appendix, we describe in details the model architectures of our experiments from section 5. As before, $l_{dim}$ is the latent dimension and $i_{dim}$ is the intermediate dimension.

We also report figures displaying the architectures of our experiment (all these figures were obtained with Netron (Roader)).

### A.1   Channel conditioning architectures

The discriminator is defined as five consecutive convolutional layers. Each convolutional layer $i$ consists of the following five steps:

1. batch normalization (except for the initial layer)

2. concatenation of the input tensor and the embedding of the condition $L_i(E(c))$ (passed through a hyperbolic tangent activation) along the channel dimension

3. convolutional layer

4. dropout layer with rate 0.15

5. activation function (LeakyReLU with slope 0.2 except for the last layer which uses the sigmoid function)

Table 4 describes the specific parameters of each convolutional layer.

Table 4: Convolutional layer parameters for discriminator with channel conditioning

| Layer | Input size | Output size | Kernel size | Stride | Padding |
|---|---|---|---|---|---|
| 1 | $(1, 28, 28)$ | $(l_{dim}, 14, 14)$ | $(4, 4)$ | 2 | 1 |
| 2 | $(l_{dim}, 14, 14)$ | $(l_{dim} * 2, 7, 7)$ | $(4, 4)$ | 2 | 1 |
| 3 | $(l_{dim} * 2, 7, 7)$ | $(l_{dim} * 4, 4, 4)$ | $(3, 3)$ | 2 | 1 |
| 4 | $(l_{dim} * 4, 4, 4)$ | $(l_{dim} * 8, 2, 2)$ | $(4, 4)$ | 2 | 1 |
| 5 | $(l_{dim} * 8, 2, 2)$ | $(i_{dim}, 1, 1)$ | $(2, 2)$ | 1 | 0 |

The generator is defined as five consecutive transpose convolutional layers. Each transpose convolutional layer consists of the following five steps:

1. batch normalization

2. concatenation of the input tensor and the embedding of the condition, along the channel dimension

3. transpose convolutional layer

4. dropout layer with rate 0.15 (except the last layer)

5. activation function (ReLU except for the last layer which uses the hyperbolic tangent function)

Table 5 describes the specific parameters of each transpose convolutional layer.

### A.2   Filter conditioning architectures

The discriminator is defined as five consecutive 'convolutional' layers. Each convolutional layer consists of the following five steps:

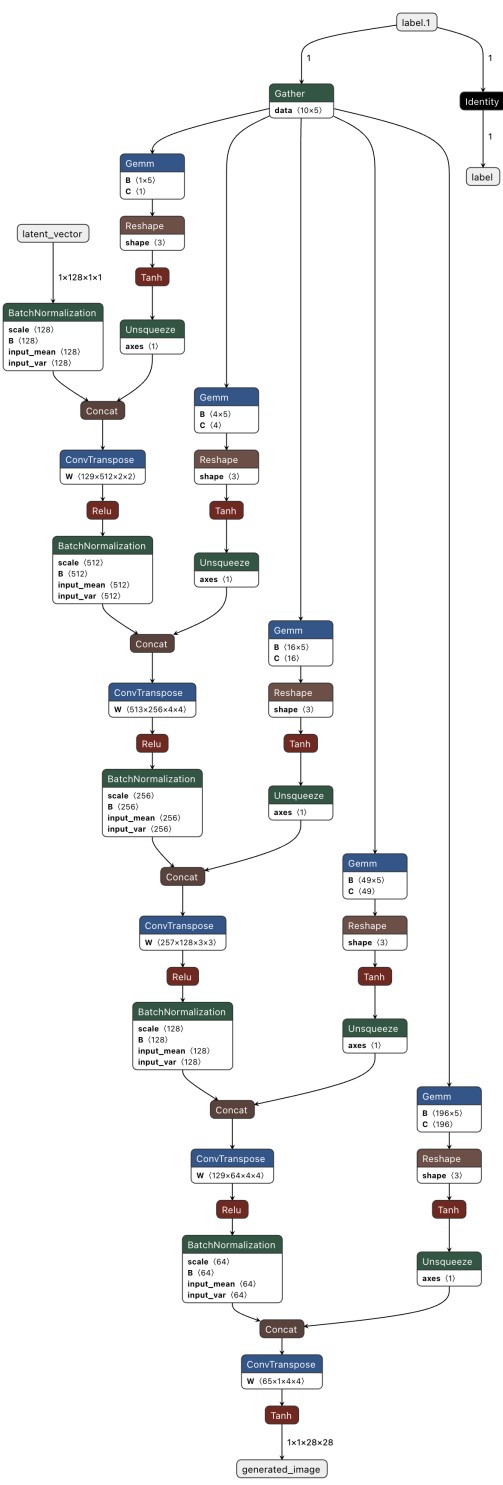

Figure 1: Architecture of the generator with channel conditioning

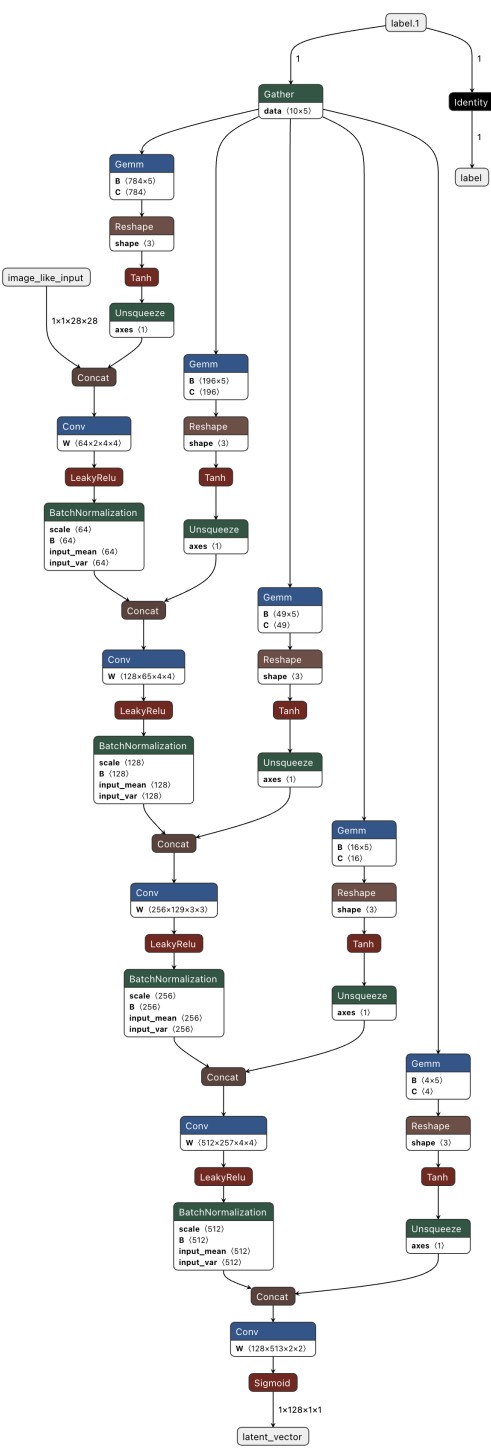

Figure 2: Architecture of the discriminator with channel conditioning

Table 5: Convolutional layer parameters for generator with channel conditioning

| Layer | Input size | Output size | Kernel size | Stride | Padding |
|---|---|---|---|---|---|
| 1 | $(i_{dim}, 1, 1)$ | $(l_{dim} * 8, 2, 2)$ | $(2, 2)$ | 1 | 0 |
| 2 | $(l_{dim} * 8, 2, 2)$ | $(l_{dim} * 4, 4, 4)$ | $(4, 4)$ | 2 | 1 |
| 3 | $(l_{dim} * 4, 4, 4)$ | $(l_{dim} * 2, 7, 7)$ | $(3, 3)$ | 2 | 1 |
| 4 | $(l_{dim} * 2, 7, 7)$ | $(l_{dim}, 14, 14)$ | $(4, 4)$ | 2 | 1 |
| 5 | $(l_{dim}, 14, 14)$ | $(1, 28, 28)$ | $(4, 4)$ | 2 | 1 |

1. batch normalization (except for the initial layer)

2. channel-wise cross-correlation between the input tensor $x_i$ and the embedding of the condition $L_i(E(c))$ (passed through a hyperbolic tangent activation)

3. channel mixer layer $C_i$, applied on the channel dimension

4. dropout layer with rate 0.15

5. activation function (LeakyReLU with slope 0.2 except for the last layer which uses the sigmoid function)

Table 6 describes the specific parameters of each convolutional layer.

Table 6: Convolutional layer parameters for discriminator with filter conditioning

| Layer | Input size | Kernel $L_i(E(c))$ size | Stride | Padding |
|---|---|---|---|---|
| 1 | $(1, 28, 28)$ | $(1, 4, 4)$ | 2 | 1 |
| 2 | $(l_{dim}, 14, 14)$ | $(l_{dim}, 4, 4)$ | 2 | 1 |
| 3 | $(l_{dim} * 2, 7, 7)$ | $(l_{dim} * 2, 3, 3)$ | 2 | 1 |
| 4 | $(l_{dim} * 4, 4, 4)$ | $(l_{dim} * 4, 4, 4)$ | 2 | 1 |
| 5 | $(l_{dim} * 8, 2, 2)$ | $(l_{dim} * 8, 2, 2)$ | 1 | 0 |

The generator is defined as five consecutive 'transpose convolutional' layers. Each transpose convolutional layer consists of the following five steps:

1. batch normalization

2. channel-wise transpose cross-correlation between the input tensor $x_i$ and the embedding of the condition $L_i(E(c))$ (passed through a hyperbolic tangent activation)

3. channel mixer layer $C_i$, applied on the channel dimension

4. dropout layer with rate 0.15 (except the last layer)

5. activation function (ReLU except for the last layer which uses the hyperbolic tangent function)

Table 7 describes the specific parameters of each transpose convolutional layer.

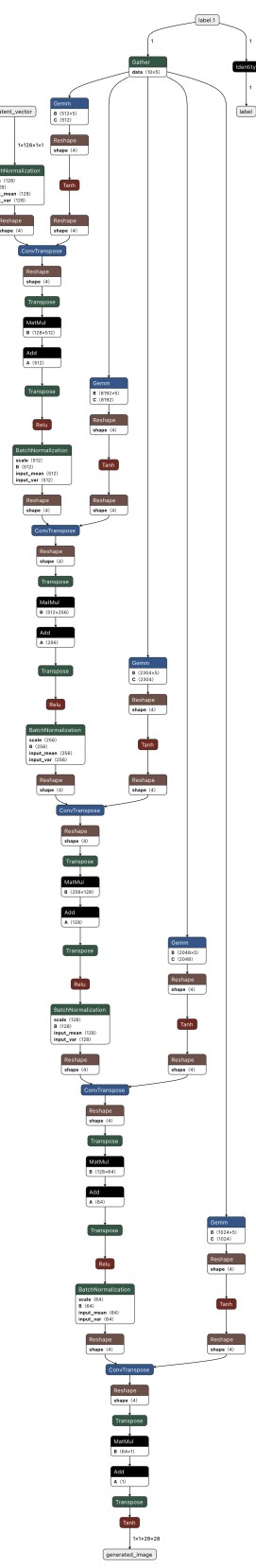

Figure 3: Architecture of the generator with filter conditioning

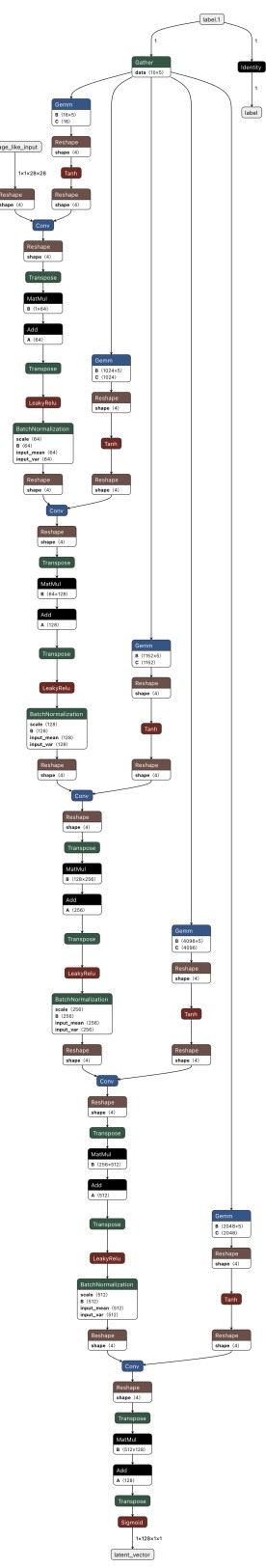

Figure 4: Architecture of the discriminator with filter conditioning

Table 7: Convolutional layer parameters for generator with filter conditioning

| Layer | Input size | Kernel $L_i(E(c))$ size | Stride | Padding |
|---|---|---|---|---|
| 1 | $(i_{dim}, 1, 1)$ | $(i_{dim}, 2, 2)$ | 1 | 0 |
| 2 | $(l_{dim} * 8, 2, 2)$ | $(l_{dim} * 8, 4, 4)$ | 2 | 1 |
| 3 | $(l_{dim} * 4, 4, 4)$ | $(l_{dim} * 4, 3, 3)$ | 2 | 1 |
| 4 | $(l_{dim} * 2, 7, 7)$ | $(l_{dim} * 2, 4, 4)$ | 2 | 1 |
| 5 | $(l_{dim}, 14, 14)$ | $(l_{dim}, 4, 4)$ | 2 | 1 |

# B   Generated images

In this section, we display synthetic images generated by our experiments on the MNIST dataset. Each group of image is generated by one experiment, starting from the same noisy sample but with different conditions.

## B.1   Channel conditioning experiments

The following images were generated by the best experiments trained with the channel conditioning mechanism described in section 5.3.

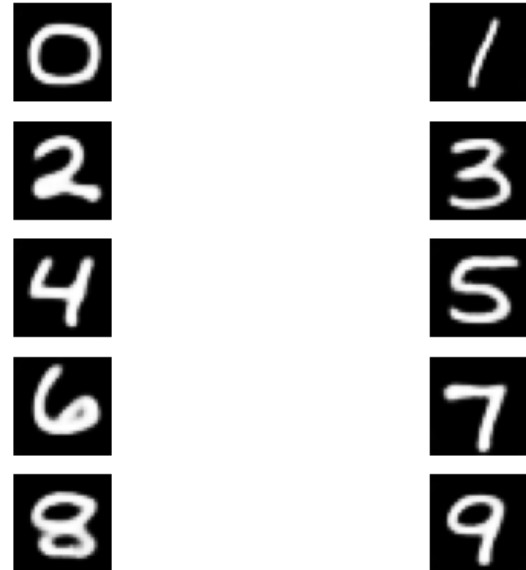

Figure 5: Images generated by small model with channel conditioning

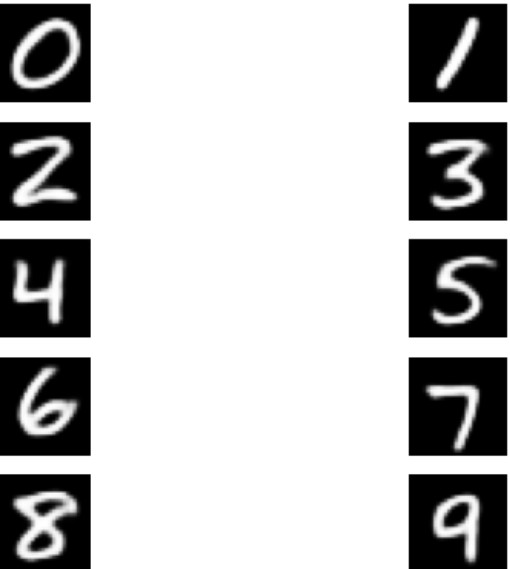

Figure 6: Images generated by large model with channel conditioning

### B.2 Filter conditioning experiments

The following images were generated by the best experiments trained with the filter conditioning mechanism described in section 5.4.

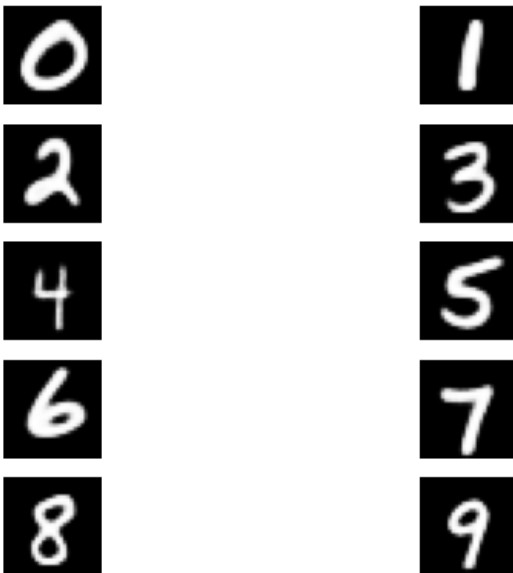

Figure 7: Images generated by small model with filter conditioning

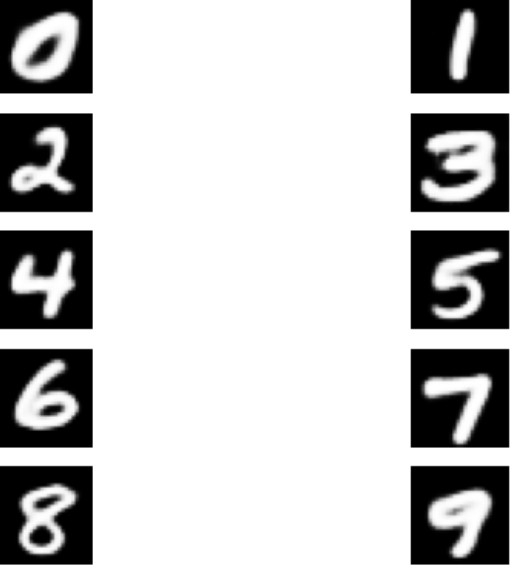

Figure 8: Images generated by large model with filter conditioning

