# OpenReview forum: "Conditional Idempotent Generative Networks"
_TMLR — Rejected by TMLR_

### Review · Reviewer_9WSU · 2024-07-28

**Summary Of Contributions:**

The authors propose a couple of implementations of a conditional version of the idempotent generative networks. The ideas are examined with MNIST images.

**Audience:**

Yes

**Claims And Evidence:**

No

**Requested Changes:**

(1)
Add a few, at least one, more experiments using different datasets. As this is *not* a theory paper, empirical investigation of the validity and effectiveness with multiple datasets, **especially with something more complex than MNIST**, is essential as a modern ML research paper.

(2)
Consider adding any baseline methods for comparison. The proposed method does **not** need to beat baselines, but please clarify that the method achieves something meaningful at least. Some visual examples are in Appendix B, but any quantitative evaluation will be highly helpful.

**Strengths And Weaknesses:**

## Strengths

+ The motivation to make IGNs conditional is clearly presented.

+ The proposed loss function makes sense.

+ The writing is okay.

## Weaknesses

- No meaningful real use case is demonstrated. MNIST is fine as a sanity check but is nothing more.

- Even the current MNIST results do not make much sense, mainly because there is no baseline at all. From what is reported in Table 3, we cannot know what kind of improvement was made by the proposed method.

Below are minor points.

- The authors repeatedly say "theoretical," but I find no theories in the paper. What is presented now rather seems to be just methodological.

- $D_c \times \hat{c}$ in Section 3.2 (and other occasions too) should instead be something like $D_c \times$ { $\hat{c}$ }?

- In Page 4,
> noisy samples ($z$, $c$) where $z$ is a randomly sampled from $X$

isn't $c$ randomly sampled too ?

- In Section 3.3,
> $C$ is a continuous manifold (for example, $c$ can be any English sentence)

In this case $C$ is not continuous and is just combinatorially large.

---

### Review · Reviewer_saS8 · 2024-08-05

**Summary Of Contributions:**

The authors propose conditioning of Idempotent Generative Networks for guided generation and demonstrate feasibility of their approach on experiments with MNIST.

**Audience:**

No

**Claims And Evidence:**

No

**Requested Changes:**

Listed in Weaknesses

**Strengths And Weaknesses:**

Strengths:

- The paper is easy to follow.

Weaknesses:

- Contributions:
  - The paper is very limited in terms of its contributions. When the authors claim to theoretical foundations for Conditional IGNs, I expect a profound theorem or result to shed light on the specific nature of conditioning arising from the idempotency. However, I found no theoretical perspective of that ilk. The implementation of conditioning is also based on standard techniques in generative modelling, except perhaps the design of the loss function. In addition to this, the experimental section is very sparse as well, in terms of scope. In light of this, I find this work lacking in terms of making significant contributions and the authors claims somewhat exaggerated.

- Exposition & Presentation:
  - The mathematical exposition is very imprecise at times; for instance, in section 2, D is referred to as both the data manifold and a probability distribution. Likewise, the loss function is not clear as the word noisy samples is used to describe both noise z as well as well as mismatched samples, d, \hat{c}.
- The writing would be benefitted by conciseness and more clarity, for instance, sec 3.3 reads like a literature survey on metrics and can be replaced by mentioning the metrics considered directly in the experiments section. Likewise, with the general exposition on channel and filter conditioning.

- Experiments:
  - The scope of experiments is very limited in terms of only relying on the MNIST dataset; I empathize with the compute limitations on the authors part but in light of the other deficiencies, the lack of further experimentation, in terms of larger networks/datasets and ablations on the loss function choices or baselines against the unconditional method, feels like a glaring omission.

---

### Review · Reviewer_DZQ1 · 2024-08-20

**Summary Of Contributions:**

The authors introduce Conditional Idempotent Generative Networks (CIGN), an extension of Idempotent Generative Networks (IGN) that incorporates a conditioning mechanism for controlled data generation. It proposes two architectures—Channel Conditioning and Filter Conditioning—to integrate conditional information and provides a theoretical foundation for these networks. The authors validate their approach through experiments on the MNIST dataset, demonstrating the efficacy of CIGNs in generating diverse images based on specific conditions.

**Audience:**

Yes

**Claims And Evidence:**

No

**Requested Changes:**

- Include a more comprehensive literature review in the introduction and related work sections. This should cover a broader range of studies on conditional generative models, such as Conditional GANs, Conditional VAEs, and other relevant architectures.

- Conduct experiments on more complex and diverse datasets beyond MNIST, such as CIFAR-10, CelebA, or other datasets that better demonstrate the robustness and scalability of the proposed CIGN architecture.

- Improve the clarity of the theoretical explanations, particularly regarding the conditioning architectures (Channel Conditioning and Filter Conditioning). Consider adding more diagrams or visual aids to help readers better understand these concepts.

**Strengths And Weaknesses:**

Strengths:
- The paper introduces an extension to the IGN by adding a conditional mechanism.

Weaknesses:
- The submission lacks the depth and breadth expected of a rigorous academic paper, as evidenced by the minimal citation of related work. Only one paper is cited in the introduction section, which indicates a limited engagement with the existing literature.

---

### Decision · Action_Editor_854F · 2024-09-19

**Recommendation:** Reject

**Comment:**

The reviewers agree that the paper is not ready for publication.

**Audience:**

No, the experiments and theoretical claims do not seem to match contemporary papers on top venues.

**Claims And Evidence:**

The paper concerns conditional generation. The contribution is to propose a network called Conditional Idempotent Generative Networks (CIGN). Both the experiments and  the theoretical benefits (as claimed in the paper) seem to be very limited and therefore I do not believe the claims are supported.